# Hybrid Technique in Temporomandibular Joint Ankylosis Arthroplasty Using Surgical Cement and Screw Fixation with Three-Dimensional Printing Planning

**DOI:** 10.3390/cmtr18020026

**Published:** 2025-04-24

**Authors:** Guilherme Pivatto Louzada, Bianca de Fatima Borim Pulino, Camila Cerantula, Gustavo Câmara, Ana Beatriz Goettnauer de Cerqueira, Gines Alves, Guilherme Zanovelli Silva, Thiago Nunes Palhares, Wendell Fernando Uguetto, Raphael Capelli Guerra

**Affiliations:** 1Hospital Sírio-Libanês, Instituto de Ensino e Pesquisa, Sao Paulo 01308-060, Brazil; drguilhermelouzada@gmail.com (G.P.L.); biancapulino@icloud.com (B.d.F.B.P.); gustavoqcamara@gmail.com (G.C.); wendelluguetto@yahoo.com.br (W.F.U.); 2Brazilian Navy, Odontoclínica Central Rio de Janeiro, Rio de Janeiro 20091-000, Brazil; 3Hospital Municipalizado Adão Pereira Nunes, Rio de Janeiro 25213-020, Brazil; anabgoett@gmail.com (A.B.G.d.C.); ginesmaxilofacial@gmail.com (G.A.); gzanovelli.bmf@gmail.com (G.Z.S.); 4Department of Oral and Maxillofacial Surgery, Hospital Leforte, Rede Américas, Sao Paulo 01507-000, Brazil; 5Department of Oral and Maxillofacial Surgery, Hospital Israelita Albert Einstein, Sao Paulo 05652-900, Brazil; 6Department of Diagnosis and Surgery, Araçatuba School of Dentistry, Sao Paulo State University (UNESP), Sao Paulo 16018-805, Brazil; camila_cerantula@hotmail.com; 7Hospital Santa Teresa, Petropolis 25680-003, Brazil; 8Centro de Tecnologia da Informação Renato Archer, Campinas 13069-901, Brazil; thiagonup@gmail.com

**Keywords:** TMJ ankylosis, hybrid technique, virtual planning, 3D printing, surgical cement, biconvex arthroplasty, PMMA, computed tomographic angiography

## Abstract

Temporomandibular joint (TMJ) ankylosis compromises essential functions such as chewing, phonation, and breathing. Surgical treatment aims to restore mandibular mobility and prevent the recurrence of joint fusion. This article describes a technical variation based on Puricelli biconvex arthroplasty, using surgical cement, screw fixation, and 3D-printed cutting guides based on virtual planning, allowing for greater precision in joint reconstruction. In this work, we present the step-by-step process used in the customization of cutting guides, virtual planning, and the production of the interposition material with PMMA associated with fixation with titanium screws as a hybrid method for the treatment of recurrent TMJ ankylosis. This reported technique is demonstrated to be reproducible, low-cost, and effective.

## 1. Introduction

TMJ ankylosis results from the pathological fusion of the mandibular condyle to the temporal bone and is classified as fibrous, osseous, or fibro-osseous [1]. Childhood trauma is a predominant etiological factor, with inadequately treated condylar fractures leading to the progressive formation of an ankylotic bone block [2]. Surgical treatment aims to restore mandibular mobility, preventing recurrences [3]. Among the available techniques, Puricelli biconvex arthroplasty stands out for creating convex articular surfaces that minimize friction, promoting better post-operative functionality [4].

The incorporation of virtual planning and 3D printing enhances this technique, allowing for the prior fabrication of customized cutting guides and anatomical molds for cementing the biconvex prosthesis. This approach increases surgical precision, reduces the operative time, and improves outcome predictability. In addition, using mini-screw fixation increases the stability of the surgical cement, reducing the risk of displacement or unwanted remodeling.

This study describes an in-house low-cost technical variation using surgical cement for condylar replacement, screw fixation, and cutting guides produced in-house using 3D printing. It also discusses the feasibility of a hybrid approach, combining surgical cement with alternative biomaterials to optimize joint rehabilitation.

## 2. Materials and Methods

This study was submitted for approval to the Ethics Committee of the Carlos Chagas Postgraduate Medical Institute (79102024.4.0000.0251).

### 2.1. Treatment Steps

Our treatment protocol will provide instructions to help healthcare professionals perform biconvex arthroplasty procedures in the treatment of TMJ bone ankylosis. The step-by-step process is divided into 3 main phases (Figure 1):

Phase 1: Computed tomographic angiography (CTA) image acquisition and virtual planning;

Phase 2: 3D fabrication of the 3D biomodel and cutting guides;

Phase 3: Intraoperative modeling, fitting, and adjustment of the PMMA prosthesis for biconvex arthroplasty.

### 2.2. Computed Tomography (CT)

Computed tomography (CT) provides a set of volume data that can be reconstructed and visualized with multiplanar transformation (MPR) tools. For CT scans of the craniofacial region, contiguous or overlapping axial slices should be acquired with a slice thickness of no more than 1 mm. In this study, CT scans were carried out using helical/spiral scanners, including Aquilion Lightning^™^ of Canon Medical Systems (Tokyo, Japan), with 32 channels from the scanner workstations. The files were saved in DICOM (Digital Imaging and Communications in Medicine) format at a 0.625 mm isotropic resolution, which is the standard format for managing medical information and related data, including CT images for later editing. After performing the CT scan and analyzing the proximity of the internal maxillary artery to the ankylotic mass in the TMJ, mainly because it was a case of re-ankylosis, we opted for an investigation using CT angiography.

### 2.3. CTA

For cranial CT angiography image acquisition, a 1.5 mL/kg body weight iodinated venous contrast of Iopamiron^®^ (iopamidol) (Bracco Imaging do Brasil Import and Distribution of Medicines Ltd., Duque de Caxias/Rio de Janeiro, Brazil) at a concentration of 350 mg iodine/mL was used as a specification, at an injection rate of 3–5 ml/sec, with tracking for the arterial phase during the first 8 s after injection. Dolphin Imaging^®^ 11.95 software was used to visualize the computed tomography slices for a better anatomical understanding. After using angiotomography, it was possible to measure a distance of 1.70 mm that the maxillary artery was crossing from the medial portions of the ankylotic mass on the right side and 3.4 mm on the left side (Figure 2A–C).

### 2.4. Three-Dimensional Image Conversion

The image dataset of the CT scan area is a series of 2D images that need to be segmented to allow the separation of bone from other tissues and converted into a 3D mesh to generate the volumetric reconstruction. This way, the DICOM file is transferred to an additive manufacturing unit. The image segmentation process can be carried out in different ways, either manually or automatically, with specific filters using the radiodensity of different tissues to select a range of pixels, which highlights and separates specific structures in the image, making it easier to identify bone from other tissues. Therefore, the 3D geometry of the cranium was obtained by rendering images via InVesalius 3^®^. InVesalius 3^®^ is an in-house, free, open-source 3D medical imaging reconstruction software that generates a 3D image from a sequence of 2D DICOM images. It was developed by the Renato Archer Information Technology Center (CTI-RA), Campinas/São Paulo, Brazil, supporting 24 languages. The software is compatible with Windows, Linux, and macOS. The use of InVesalius^®^ 3.1v software has been scientifically verified to produce 3D biomodels and prototypes comparable to those produced by the MIMICS 17.0v software (Abdullah., 2019) [5]. This has made it an excellent option for research institutions and public hospitals in low-income countries to produce accurate STL skull models for teaching and medical training purposes.

### 2.5. Virtual Planning and Making the Cutting Guide

The next stage involved segmenting the CT file (DICOM) and converting it into a stereolithography file (.STL), which allowed the osteotomy guides to be manipulated and made. The virtual surgical planning for making the cutting guide was carried out in an interactive online meeting with the clinical engineers at CTI Renato Archer and the surgeon. For this purpose, the 3D modeling of the cutting guides was carried out using the Autodesk Meshmixer V 3.5 software. During this process, the distance specifications and access angles determined by the surgeon were considered, ensuring a personalized approach to the development of the guides. According to the anatomical position of the ankylotic mass and the maxillary artery, we opted for an osteotomy in the mandibular ramus located more inferiorly in the cranio-caudal direction, as it is an area with a lower bone thickness and facilitates trans-operative visualization. To this end, a 30 mm gap was planned for the cutting guide, with the distance between the maxillary artery and the ankylotic mass as the anatomical reference point (Figure 3). It was possible to print the cutting guides and the patient’s skull using a 3D printer from this generated file (.STL) (Figure 4). In this case, we used the 3D Systems 3D printer (model: Sinterstation^®^ HiQ^™^ series SLS^®^ Systems (San Diego, CA, USA)), using the selective laser sintering technique (SLS^™^) with a nylon-based printing material (PA12).

### 2.6. Three-Dimensional Printing

The biomodels and cutting guides were manufactured using the SLS (selective laser sintering) 3D printing technique, using polyamide 12 (PA12) as the base material. This printing technique provides high precision and quality of detail. PA12 offers good thermal resistance for sterilization in an autoclave and physical resistance, guaranteeing the functionality of the cutting guides in the clinical context. The printing process was carried out with a layer height of 0.1 mm and a CO₂ laser power of 15 W, ensuring the precise control of the material fusion and dimensional fidelity of the printed parts.

### 2.7. Sterilization Process

PMMA guides are printed and must undergo a sterilization process to ensure that the handling of the material during the trans-operative phase does not cause any contamination. It is, therefore, recommended that the guides undergo a sterilization process. The product can be sterilized using the autoclave method, which uses saturated steam at 121 °C. In our protocol, it was recommended to follow ISO 17665-1: which deals with the sterilization of health products and states that it takes about 15 to 30 min under a pressure of 106 kPa (1 atm) to reach 121 °C. The time should be marked when the heat in the sterilization chamber reaches the desired temperature. Some polymers, including PA, can be safely sterilized with steam at 121 °C for up to 1 h or at 134 °C for 5 min.

### 2.8. Surgical Procedure Sample

The surgical treatment was performed under general anesthesia with nasotracheal intubation using an Olympus^®^ BF-1T30 flexible bronchoscope (HP Medical (São Paulo, SP/Brazil)). The retromandibular access with Risdon submandibular extension was used to access the ankylotic mass and allow the cutting guide to be placed (Figure 5A). The guides were stabilized with two 1.5 mm NEOORTHO^®^ Produtos Ortopédicos S/A (Curitiba, PR/Brazil) screws. The osteotomies were performed with a reciprocating micro-saw (SR016 Razek^®^ (São Carlos, SP/Brazil)), and a 30 mm gap was made (Figure 5B). After the osteotomies, the ankylotic block was separated using a chisel and hammer. On the right side, as the temporalis fascia flap interposition technique had already been performed, it was planned to make biconvex spheres using high-viscosity Osteo-Class^®^ acrylic bone cement composed of methyl/polymethyl methacrylate and a radiopaque agent (Figure 6). The material was handled outside the surgical site and adapted to the 3D-printed biomodels. After it had set and was in the final stages of hardening, it was adapted to the bone structure on each side of the bone stumps, with a 0.9% saline solution to prevent the bone surface from overheating. At the end, the material was stabilized using two 1.5mm NEOORTHO^®^ Produtos Ortopédicos S/A (Curitiba, PR/Brazil) screws in each portion of the bone remnants (Figure 7). On the left side, the temporalis fascia flap was rotated over the zygomatic arch and stabilized with 3.0 polypropylene thread on the surface of the zygomatic arch. As the coronoid process was elongated on this side, it was osteotomized to avoid a larger area of mechanical locking and potential new ankylosis in the region. An opening of 35 mm was achieved during the trans-operative period.

### 2.9. Manufacture of Surgical Cement Spheres and Installation

High-viscosity Osteo-Class^®^ acrylic bone cement (Baumer S.A, São Paulo, Brazil) is a medical product made up of an implantable, self-curing, and radiopaque acrylic compound developed for use in orthopedic and dental surgeries, meeting the regulatory country requirements. High-viscosity acrylic bone cement comprises the following mixing components: the polymer envelope (fast polymethyl methacrylate); slow polymethyl methacrylate; butyl polymethacrylate; and zirconium oxide. The mixing component, polymethyl butyl methacrylate, is used to initiate the polymerization of the acrylic cement. (Monomer ampoule: monomethyl methacrylate; *N*,*N* dimethyl-p-toluidine; and hydroquinone, USP). The liquid is added to the powder in a stainless-steel bowl and mixed for 3 min with a spatula until the powder is completely saturated with the liquid to minimize air entrapment. The handling point is when the mass no longer clings to the glove with few alloys.

Two separate spheres of surgical cement are then manipulated by hand, first onto the 3D biomodel, and then transferred for insertion into the bone segment. At this point, it is recommended that the pieces be refrigerated with a 0.9% saline solution, as an exothermic reaction will be initiated, which will continue throughout the adaptation phase until the material is fixed in position and can no longer be removed. During the adaptation of the material, it is recommended that some type of retention be made on the medial and lateral faces of the mandibular ramus, prolonging the shape of the material and allowing a surface for the installation of two screws on the lateral face of the bone.

## 3. Post-Operative Rehabilitation and Follow-Up

The patient had phonation difficulties in the first few weeks. Therefore, he began to be monitored from the third week after the operation by a phonologist, who continued to visit him every week for the first two months. The patient was already achieving better diction and speech at the end of this period. Another difficulty was muscle limitation due to trismus and atrophy of the muscles. This even led to a spontaneous reduction in mouth opening during the first month, which can be correlated with post-operative edema and surgical trauma. To this end, a weekly physiotherapy protocol was introduced from the immediate post-operative period, with three weekly meetings during the first month, decreasing to twice a week in the following month and once a week in the third month, until the end of the physiotherapy follow-up. The treatment was based on myofascial release maneuvers and thermotherapy (local heat for 10 min) and muscle mobility exercises for excursive and lateral movements and maximum mouth opening. Daily exercises were taught to the patient to perform manual stretching in front of the mirror. The diet was pasty during the first 3 weeks and changed to a more consistent one after the end of the fourth week. As it was possible to see a clear progression in improvement over the weeks, and the patient showed positive progression after each session, it was possible to diagnose that the trismus condition he presented at the beginning of each session was related to a muscular component and not a mechanical restriction (Figure 8A,B).

In addition, photobiomodulation therapy with red laser therapy (660 nm) was applied to the facial muscles, with a dose of 2 J (40 mW for 20 s) (Laser Therapy XT DMC^®^, São Carlos, Brazil) per point and an average of three points on the masseter, two on the temporalis, and two on the medial pterygoid. After this period, the patient continued to attend outpatient appointments at the hospital once every 3 months, with signs of stable development and no recurrences. By the end of the second month, the patient was able to keep his mouth open more than 30 mm without forcing and more than 30 mm while forcing. Furthermore, a control CT scan was carried out 8 months after the operation, which showed that the space of the surgical cement had been maintained, with no displacement of the material, as seen in the control CT scan 16 months after the operation. During the last follow-up visit, the patient showed great satisfaction with the progress of the treatment and could even consider starting oral rehabilitation treatments and resuming social activities.

## 4. Discussion

Authors have consistently reported that the TMJ can be affected by many disease processes resulting from developmental disorders, neoplasms, trauma, arthritic diseases, previous unsuccessful joint surgery, or fibrous or bony ankylosis [1,2,3,4,6]. In addition, the functional deficit caused by limitations in mandibular movement leads to a series of nutritional and developmental changes [2,6]. The occurrence of this event during childhood and the age of bone development leads to significant changes in mandibular growth, resulting in dentofacial deformity, as seen in this case, where the patient developed mandibular micrognathia [6]. This can be explained because the mandibular condyle is an important growth center for the maxillofacial bones. Therefore, especially in patients with a history of intracapsular or comminuted fractures who were treated with excessive immobilization time, a marked increase in bone mineralization and healing of the condylar region is observed, which leads to condylar hypofunction. This results in changes in the development pattern of the entire maxillomandibular complex and in those who experience ankylosis early on [1,2,4,5,6,7]. In the case in question, the patient had a history of falling off a bicycle in childhood, which was treated with a prolonged maxillomandibular block. The patient spent almost two months with the block, keeping his mouth closed due to a lack of knowledge and difficulty in accessing information. The lack of function attributed by the patient was not correlated with a progressive limitation of his functions, which, later, led to the underdevelopment of the maxillary bones, with a significant mandibular retrusion that affected several social aspects.

Puricelli (2022) described that the need for surgical removal of TMJ structures creates spaces between the mandible and the base of the skull, which can be reconstructed using cartilaginous tissue engineering or alloplastic prostheses [7,8,9,10]. Techniques for reconstructing articular structures, which are still being researched, can include the condyle or articular fossa alone or a combination of them in a total prosthesis [8]. Different biomaterials, such as chromium (Cr) and cobalt (Co) alloys, titanium, and ultra-high-molecular-weight dense polyethylene can be used for TMJ prosthetic reconstruction. Total prostheses are more biomechanical than a biological solution for the treatment of severe joint diseases, such as ankylosis [4,9]. The advantage of using customized prostheses or prostheses for joint reconstruction over osteogenic distraction techniques, or the use of autogenous costochondral grafts, is lower surgical morbidity, greater treatment stability, and the possibility of resolution in a single operative stage [4]. Wolford et al. (2003) stated that an articular prosthesis is more effective to manage in the trans-operative period with a distorted and mutilated anatomy, without depending on periarticular vascularization, unlike in the case of autogenous grafts, such as costochondral and sternocleidoclavicular grafts, where local vascularization is important for the maintenance of the graft and the rate of resorption [4,8]. However, this is a very remote reality in terms of the public health service in Brazil.

Gap arthroplasty without interposition requires a greater amount of bone resection than arthroplasty with interposition [3,10]. According to Topazian (2001), 53% of patients develop recurrence of ankylosis, regardless of the size of the gap, and intensive physiotherapy in the immediate post-operative period [11]. A multitude of treatments are used to preserve the maintenance of mouth opening and mandibular functions without recurrence, since limitations in the amplitude of mouth opening, generally after 6 months of surgery, indicate a greater potential for re-ankylosis [12]. There is no consensus that a single method is consistently the gold standard for successful treatment [9,12]. However, according to Topazian (2001), the regularity of post-operative physiotherapy and the care taken by the patient to comply with the strict maintenance of joint mobility exercises in the post-operative period are closely related to the likelihood of re-ankylosis [11].

For this technique, a wide variety of autogenous materials, such as auricular cartilage, costochondral grafts, and temporal muscle fascia, as well as alloplastic materials, such as silicone, PMMA, and metals, can be used [3,7,10,11,13,14]. Polymethylmethacrylate (PMMA)-based surgical cement is used in various procedures, such as hip and knee prostheses and cranioplasty [7,13,15]. Changes to the structure of the bone using cement are introduced to restore the functionality of the TMJ, always considering its thermal, mechanical, and biological properties to seek better performance. With new characteristics, multifunctional cements are efficient alternatives in the manufacture of prosthetic devices. This was demonstrated in a case report by Puricelli et al. (1997) [14], who found that good results can be achieved with follow-ups of more than 20 years with the use of cement in TMJ surgery. In the present case report, it was observed that the treatment of right TMJ arthroplasty with temporalis fascia evolved into re-ankylosis years later. Given the financial limitations, because it was not feasible to carry out joint reconstruction using a customized prosthesis, or even to use autogenous resources for interposition, we opted to use the biconvex arthroplasty technique using the high-viscosity Osteo-Class^®^ acrylic bone cement, composed of methyl/polymethyl methacrylate and a radiopaque agent, because it is a low-cost technique with good stability of results and is an alternative for cases of re-ankylosis in the reality of healthcare conditions for the poor in Brazil [14,16,17].

Most current biomaterials are well tolerated by the body. In addition, these biomaterials maintain structural integrity, achieve mechanical stability in the bone, and are not colonized by microorganisms. Acrylic bone cement based on polymeric ceramic composites in methyl methacrylate (PMMA) has been widely used in joint replacements for decades [13]. Puricelli’s biconvex arthroplasty technique allows reconstruction using two convex PMMA surfaces [14]. Puricelli (1997) [14] explained that in the biconvex arthroplasty approach, a prefabricated self-curing methyl methacrylate prosthesis is introduced, seeking anatomical reconstruction of the TMJ alongside biomechanical improvement. The choice of two convex surfaces reduces friction between the areas, facilitating muscle movement with less force during the displacement of the condyle. The intentional modification of the force vector from an anteroposterior/superoinferior to a posteroinferior/superoinferior direction is supported by the corresponding bone remnant of the roof of the glenoid fossa and the posterior wall. This modification allows the correction of mandibular retrusion and ramus height to be maintained, ensuring stability over time. In the present case, it was not possible to observe significant sagittal and vertical changes from a skeletal point of view due to severe mandibular micrognathism. For this case, a second surgical correction with bimaxillary advancement and rotation of the occlusal plane would be indicated [14,16,17].

The observational post-operative period has confirmed the efficacy of this technique since 1978, demonstrating the stability and absence of pain in the region after the surgical procedure, which showed the ability of the alloplastic material to absorb forces and restore functionality. Mastery in handling methyl methacrylate, combined with academic knowledge in the dentistry field, allows the oral and maxillofacial surgeon to obtain the best results with this material, as it is commonly used during their training [16]. Technically, the setting time associated with the heating of the surgical cement is the major challenge in allowing the material to adapt correctly to the surgical site. Prior manipulation and adaptation of the material in a 3D biomodel provides a better finish and fit. In this case, the use of a titanium screw to allow the surgical bone cement to be fixed in the desired position in the bone stump and to guarantee its stability was chosen in the surgical procedure, unlike the technique described by Puricelli in 1997 [14]. The great advantages of the material used are the return of joint mobility in 48 h and the ease with which it can be worn with a drill and added by apposition. As such, it can replace the condyle in its perfect shape and correct the mandibular replacement and ramus height through posterior support [17].

The risk of a trans-operative complication with bleeding from the internal maxillary artery is a major concern during the removal of an ankylotic mass [18,19]. Bouloux and Perciaccante (2009) reported a case of intercurrence without the possibility of containing bleeding despite ligation of the external carotid artery, which the author explained by the presence of vessels of the opposite external carotid artery, internal carotid artery, and collaterals that maintain peripheral irrigation, which can cause significant blood loss [20]. CT angiography is an important resource for identifying bone boundaries and proximity to the branches of the maxillary artery and was used in this case. Okada et al. (1996) demonstrated that selective arterial embolization using CT angiography can reach more inaccessible areas than ligation in a safe manner [21,22]. In this case, this option was not possible and was only available in an emergency for possible ligation due to the limitations of the presence of a vascular surgeon during the operation. We opted to use the customization feature of a cutting guide to increase the precision of the technique due to the proximity of less than 2 mm to the ankylotic mass, which could be assessed using virtual planning and 3D reconstruction after angiography. Since the program used to create the guides is available in open-access software and uses the author’s printer, we opted for the in-house customization of the cutting guides, which made it safer and faster to separate the ankylotic mass, as stated by Franco et al. (2021) [23].

The post-operative recovery time depends on the extent of the surgical procedure and the time the patient remained with TMJ ankylosis. The first week is more related to recovery from the surgical treatment itself. Complete functional recovery can vary from four to twelve weeks [6,7,8,12,13,14,15,16,24]. In the present case, since the patient had a case of re-ankylosis with which he spent more than 10 years without being able to undergo any treatment, this made recovery more time-consuming. Therefore, special attention from the speech therapist was required so that he could recover his diction correctly since this long period affected the patient’s awareness of pronunciation and communication. In addition, the patient’s chewing efficiency was significantly affected, whereby the strength to grind more consistent foods was impaired, so the patient was only able to chew efficiently at the end of the first month after surgery.

## 5. Conclusions

The variation of the Puricelli technique with surgical cement, screw fixation, and virtual planning with 3D printing is an accessible and effective alternative for TMJ reconstruction. Its evolution into a hybrid technique may represent an advance in the rehabilitation of patients with re-ankylosis, especially in public health systems in low-income countries, as it is a reproducible, safe, effective, and customized in-house technique.

## Figures and Tables

**Figure 1 cmtr-18-00026-f001:**
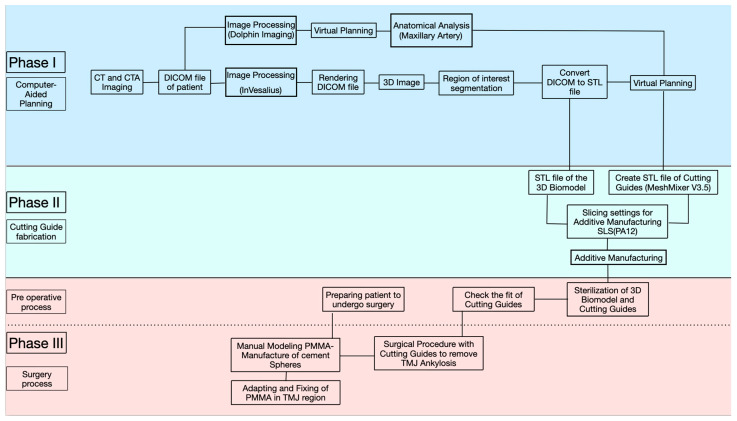
Diagram explaining the workflow adopted during the 3 phases.

**Figure 2 cmtr-18-00026-f002:**
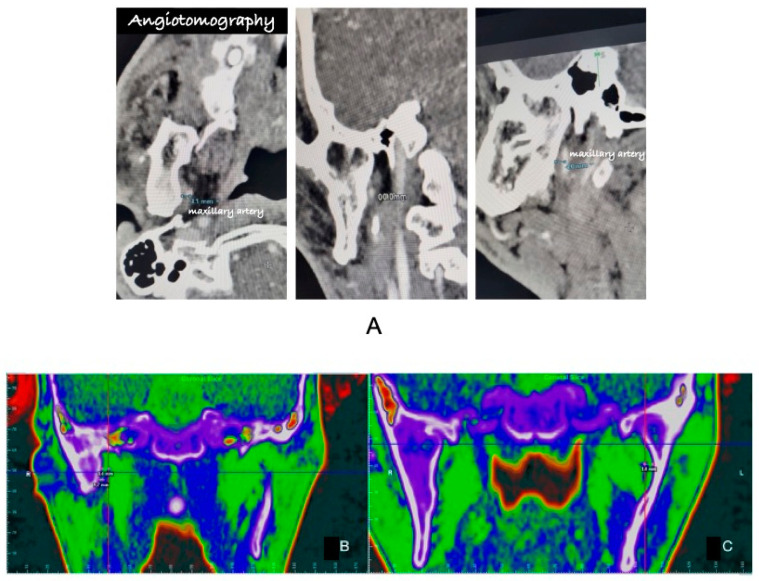
(**A**) Axial, coronal, and sagittal views of the CTA scan after iodinated contrast, respectively, showing the relationship between the right-sided ankylotic mass and the maxillary artery. (**B**) Relationship between the maxillary artery and the ankylotic mass on the right side. Color inversion map used in Dolphin Imaging Software 11.95 to evaluate the CT after angiotomography. (**C**) Relationship between the maxillary artery and the ankylotic mass on the left side. Color inversion map used in Dolphin Imaging Software 11.95 to evaluate the CT after angiotomography.

**Figure 3 cmtr-18-00026-f003:**
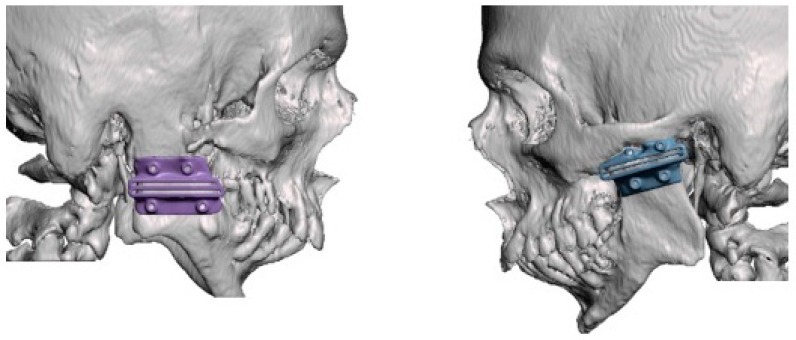
Three-dimensional planning of cutting guides placed in position to assess the relationship of the guide with the positioning of the ankylotic mass to facilitate surgical access planning. In the right condyle, we opted to place the guide in a more inferior position in relation to the mandibular ramus due to the thickness of the ankylotic mass and the positioning of the maxillary artery. This facilitated the exposure of the region for adapting the guide and, later, making the PMMA for interposition in the gap.

**Figure 4 cmtr-18-00026-f004:**
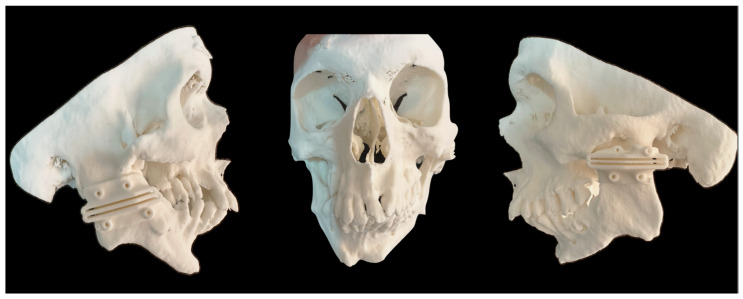
Test of the adaptation of the printed cutting guides to the biomodel after the sterilization process to ensure that there were no deformations in the surgical guide.

**Figure 5 cmtr-18-00026-f005:**
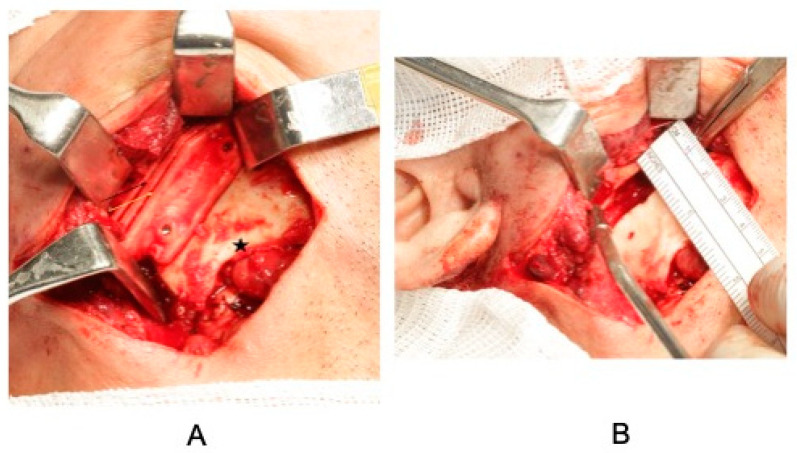
(**A**)—The asterisk corresponds to the gonial angle, which has a hypodevelopmental deformation due to the muscular action of the masseter on a mandibular ramus that suffers from a long period of ankylosis. It is possible to see the cutting guide adapted to the lateral portion of the mandibular ramus, according to the virtual planning. The black arrow shows the entrance to the reciprocating saw for marking the uppermost osteotomy, while the yellow arrow shows the lower osteotomy. (**B**)—Bone gap created in the ankylotic mass after osteotomy with the cutting guide.

**Figure 6 cmtr-18-00026-f006:**
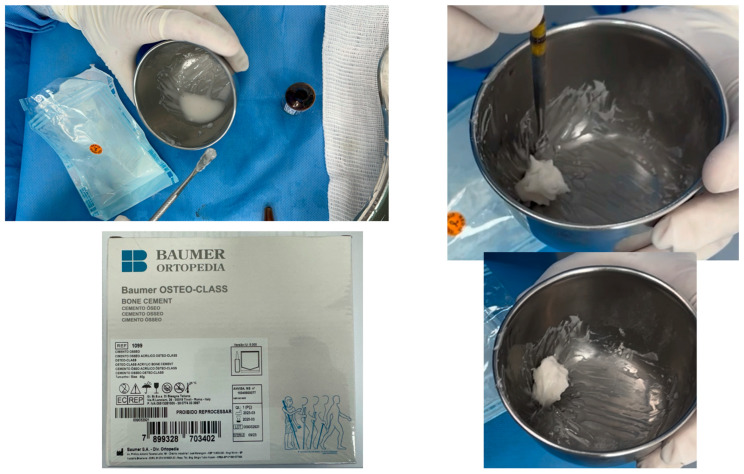
The point of manipulation of the surgical cement is when it begins to detach from the walls of the stainless-steel container, passing through the sticky phase, and acquiring greater consistency. Care should be taken not to introduce the material while it is still in its final setting phase. In addition, after adaptation in the bone cavity, copious irrigation with saline solution is important to avoid local overheating since an exothermic reaction will be observed during the material’s final setting phase.

**Figure 7 cmtr-18-00026-f007:**
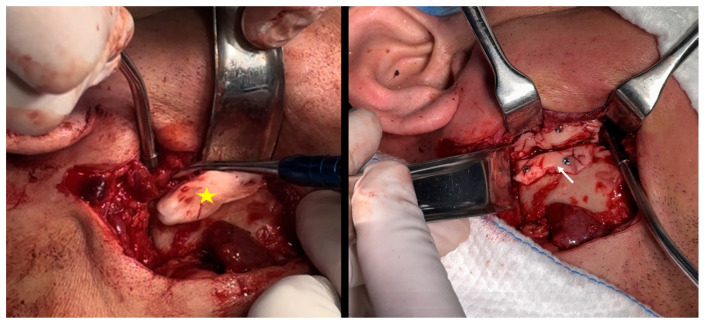
The yellow star shows the surgical cement adapted to the bone stump in the working phase, copying the entire bone structure and maintaining a plate surface along the lateral edge of the branch, to allow fixation with 2 titanium screws. The white arrow shows the caudal portion of the surgical cement adapted to the mandibular ramus. The surgical cement is fitted on each side of the bone segments and fixed with two screws on each side.

**Figure 8 cmtr-18-00026-f008:**
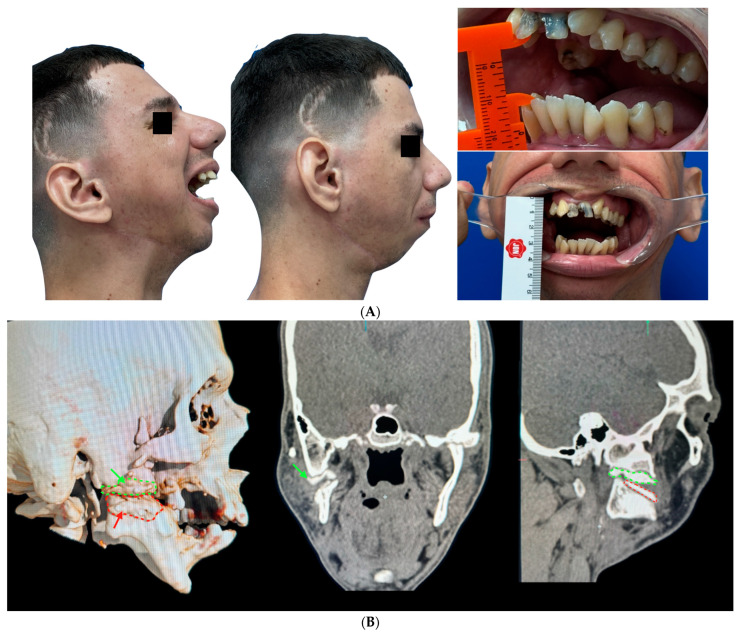
(**A**) Sixteen-month post-operative follow-up with satisfactory mouth opening. After the physiotherapy sessions, the patient was able to achieve a mouth opening of more than 30 mm, allowing him to resume his chewing and swallowing functions satisfactorily and without limitations. (**B**) A 16-month follow-up CT scan showed good adaptation of the surgical cement with no signs of TMJ re-ankylosis. In the images, the green arrow and the dotted lines outline the surgical cement in the cranial portion in the articular eminence, and the red arrow with red dotting outlines the surgical cement adapted in the caudal portion, located in the mandibular ramus.

## Data Availability

The data presented in this manuscript is not available due to privacy concerns.

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
