# Peer review of "Hybrid Technique in Temporomandibular Joint Ankylosis Arthroplasty Using Surgical Cement and Screw Fixation with Three-Dimensional Printing Planning"

_1943-3883, 2025, doi:10.3390/cmtr18020026_

Round 1

Reviewer 1 Report

Comments and Suggestions for Authors

The article in general is well written and supplemented with clear figures.

1) Title

Long and complex. The strength of this article is on the hybrid technique. The authors may consider to make it more concise and shorter. An example for consideration:

Hybrid Technique in TMJ Ankylosis Arthroplasty Using Surgical Cement and Screw Fixation.

2) Material and Methods.

Line 47: please remove 'approval number' (just maintain the numbers)

Minor typographical errors; please check and revised accordingly (eg. Line 52: Virtual to 'virtual')

Standardize the usage of term CT scan and CTA through out the manuscript.

Line 108: DICON to DICOM

Standardize the term bio-model or biomodel.

3) Heading: Results? Post-operative? 

In post-operative protocol: Authors may consider to provide more detailed information (eg. antibiotics? pain control? frequency of jaw exercise? diet? when exactly is CT scan needed postoperatively?)

Please provide information on patient pre and post-operative mouth opening.

Thank you.

Author Response

Reviewer 1

  • The title change has been considered
  • The requested grammar and standardization of terms has been reformulated
  • As this study presents a variation of the technique rather than a case report, the requested information regarding mouth opening measurement, diet, medications, and postoperative physiotherapy protocol may not be relevant. However, regarding tomographic evaluations, imaging is necessary immediately after the procedure, with follow-ups at 6 months, 1 year, and 1.5 years to ensure proper monitoring.

Reviewer 2 Report

Comments and Suggestions for Authors

Thank you very much for this excellent presentation of a technique variation for treating TMJ ankylosis. Please consider revising the title to emphasize primarily the arthroplasty technique or the treatment planning technique. Both are worth including. Please explain more regarding the cost of such manufacturing techniques. Cost is mentioned in the title, but there is no cost comparison with other techniques and no cost analysis referenced from existing literature. It is unclear whether the present study is actually "in house." Some of the treatment planning appears to have been completed in house, but there is mention of the use of engineers for virtual planning, which would seems to be an outside virtual planning source and would have its own cost. Techniques exist for completing the virtual planning completely in house, but those do not appear to have been used for this. It would be a good idea to include a table of steps for the authors and whether they are in house, out of house, and the average cost of each at the treating institution since the steps become slightly difficult to follow. This is an excellent technique overall, and the article has merit. Thank you again for your submission.

Comments on the Quality of English Language

Only minor grammatical changes are needed.

Author Response

Revisor 2

Agradecemos sinceramente aos revisores por sua análise cuidadosa de nosso manuscrito. Suas valiosas opiniões, críticas e sugestões foram essenciais para melhorar a clareza e a qualidade do trabalho final.

Reviewer 2

            Cement and PLA for 3D printing are low-cost materials. With access to a 3D printer and the use of open segmentation and planning software, it is possible to carry out the entire technique for making cutting and fixation guides for arthroplasty. With regard to costs, it is not possible to make a direct comparison between developing countries and first world countries. However, this technique appears to be more accessible, as it reduces the cost of shipping and manufacturing in specialized centers, as well as enabling local production of cutting guides, fixation guides and customization of fixation systems.

Reviewer 3 Report

Comments and Suggestions for Authors

INTRODUCTION

Give a brief and detailed description of the technique of Puricelli biconvex arthroplasty and explain the reason for improvement that needs to be developed for modifying the Puricelli technique.

SURGICAL TECHNIQUES AND METHODS

  1. As this article includes a case of TMJ ankylosis, it is important to present the case as well (not only the post-operative follow-up) to make it more understandable for the readers (the case report should be present before SURGICAL TECHNIQUES AND METHOD)
  2. Attach the surgical phases of surgery description using a diagram for better understanding
  3. Figures 1B and 1C could be put into one frame to make a clear comparison
  4. In Figure 2, images before the installation of the cutting guide photo should be included and the explanation
  5. In Figure 4, intraoperative before adaptation of the cutting guide of the surgical bed should be attached to make more understandable and describe the figure
  6. In Figure 5, the image is less focused on the surgical cement material because the hand image is too dominant
  7. In Figure 6, the images look the same and there is no explanation
  8. In Figure 7A, if possible, could be completed with other photos of clinical result and the quantitative data (in mm)
  9. In Figure 7B arrow could be added to explain which is the cement material
  10. The figures included in this article (e.g., CT scans, cutting guides, surgical cement adaptation, etc.) are informative but could be enhanced with higher resolution images or annotations to improve clarity for readers unfamiliar with medical imaging

POST-OPERATIVE REHABILITATION AND FOLLOW-UP

  1. During the follow-up, it is necessary to explain the issues or complaints from the patient that are still being experienced.
  2. Attach the development of clinical examination results in the follow-up timeline before the 16-month follow-up as the only evaluation of the 16-month follow-up described in this paper.
  3. The article briefly mentions achieving a 35mm opening during surgery but does not elaborate on long-term clinical outcomes (16-month follow-up). Besides postoperative functionality, patient satisfaction needs to be included as well. The Follow-up data or case studies would strengthen the evidence supporting the technique's efficacy

DISCUSSION

  1. In the first paragraph, an explanation about the cause and the age in which the ankylosis occurs should be added as it relates to the micrognathia complication.
  2. In the second paragraph, it will be clearer if this is divided into two paragraphs. The first paragraph explains the reconstruction techniques of articular structure that are still being researched. In addition, it needs to add the technique being developed in this operation. Regarding the osteotomy in the ramus with the linear cut, will this affect the patient's ROM?. The second paragraph related to the biomaterials used and described them in detail.
  3. In the last paragraph, the problems that still exist in the patient need to be explained, along with the possible causes and ways to address those issues. During the trans-operation period, it was stated that the mouth opening reached 35 mm, but until the 16-month post-operative follow-up , did the mouth opening get to 35 mm as well? If not, what is the possible cause?

Comments on the Quality of English Language

the paper is beat around the bush. make it simple, deep, and understandable.

Author Response

We sincerely thank the reviewers for their careful analysis of our manuscript. Their valuable opinions, criticisms, and suggestions have been essential for improving the clarity and quality of the final work.

Reviewer 3

Introduction

            While we acknowledge the significance of Puricelli’s biconvex arthroplasty, our study does not aim to provide an extensive description of this technique, as it has already been well documented in the literature. Instead, our focus is on presenting a technical variation that integrates this established method with digital planning, emphasizing its advantages in terms of cost-effectiveness, surgical precision, and predictability.

Surgical techniques and methods

  • This article presents the description of a hybrid technique, illustrated through a demonstrative case, similar to other articles recently published in the latest edition of CMTR that have adopted the same approach.
  • Use of the diagram to illustrate the workflow
  • Placed images 1B and 1C in the same frame
  • The description of image 2 is added to the subtitle in detail.
  • Picture 4 was corrected and detailed as instructed.
  • Added a new image and changed the previous one to highlight the working point of the surgical cement.
  • Correcting the image and changing the repeated image.
  • New follow up photos and the measurement in millimeters of the re-established mouth opening.
  • The structures of the surgical cement adapted to the bed are highlighted with an arrow and dotted for better illustration for the reader.

 10) Improved all images

POST-OPERATIVE REHABILITATION AND FOLLOW-UP

            Information about the post-operative period and patient management was introduced.

DISCUSSION

1-Corrections related to the patient's micrognathia and ankylosis as well as the etiology of the trauma were added.

2- Adjustments were made in the discussion and in the description of the surgical technique, highlighting the importance of mandibular ramus osteotomy in improving mouth opening amplitude.

3- Corrected highlighted paragraph.

Round 2

Reviewer 3 Report

Comments and Suggestions for Authors

The journal article titled "Hybrid Technique in Temporomandibular Joint Ankylosis Arthroplasty Using Surgical Cement and Screw Fixation with Three-Dimensional Printing Planning”

  1. Figure 1A is replaced, become Figure 1 and divided to A, B, and C, and each of the figures should give a brief description.

  1. Figure 1B is changed to Figure 2, divided A and B, and should give a brief description of each figure.

  1. Figure 2 is replaced to Figure 3 divided into A and B and should give a brief description of the figure.

  1. Figure 3 is replaced to Figure 4, divided into 3 subfigures: A, B, C, and should give a short description of the figure.

  1. Figure 4 is replaced to Figure 5 become A and B. There is already a descriptive sentence, but there are no A and B labels in the figure yet.

  1. Figure 5 is replaced, become Figure 6 consisting of A, B, C, and D and should give a short description of the figure each. Make sure all of the figures are the same size and align the left and right margins.

  1. Figure 6 is replaced to Figure 7, divided to A and B and should give a short description of each figure.

  1. Figure 7A is replaced to Figure 8, which consists of A, B, C, and D, and should give a short description of each figure.

  1. Figure 7B is replaced to Figure 9 divided into A, B, and C And should give a short description each.
  2. The conclusion should be in line with the title
